# Opportunities and Advances in Radiomics and Radiogenomics for Pediatric Medulloblastoma Tumors

**DOI:** 10.3390/diagnostics13172727

**Published:** 2023-08-22

**Authors:** Marwa Ismail, Stephen Craig, Raheel Ahmed, Peter de Blank, Pallavi Tiwari

**Affiliations:** 1Department of Radiology, University of Wisconsin-Madison, Madison, WI 53706, USA; sccraig2@wisc.edu (S.C.); ptiwari9@wisc.edu (P.T.); 2Department of Neurosurgery, School of Medicine and Public Health, University of Wisconsin-Madison, Madison, WI 53706, USA; raheel.ahmed@neurosurgery.wisc.edu; 3Department of Pediatrics, College of Medicine, University of Cincinnati, Cincinnati, OH 45267, USA; peter.deblank@cchmc.org; 4Department of Biomedical Engineering, University of Wisconsin-Madison, Madison, WI 53706, USA

**Keywords:** medulloblastoma, radiomics, risk stratification, radiogenomics, molecular subgroups

## Abstract

Recent advances in artificial intelligence have greatly impacted the field of medical imaging and vastly improved the development of computational algorithms for data analysis. In the field of pediatric neuro-oncology, radiomics, the process of obtaining high-dimensional data from radiographic images, has been recently utilized in applications including survival prognostication, molecular classification, and tumor type classification. Similarly, radiogenomics, or the integration of radiomic and genomic data, has allowed for building comprehensive computational models to better understand disease etiology. While there exist excellent review articles on radiomics and radiogenomic pipelines and their applications in adult solid tumors, in this review article, we specifically review these computational approaches in the context of pediatric medulloblastoma tumors. Based on our systematic literature research via PubMed and Google Scholar, we provide a detailed summary of a total of 15 articles that have utilized radiomic and radiogenomic analysis for survival prognostication, tumor segmentation, and molecular subgroup classification in the context of pediatric medulloblastoma. Lastly, we shed light on the current challenges with the existing approaches as well as future directions and opportunities with using these computational radiomic and radiogenomic approaches for pediatric medulloblastoma tumors.

## 1. Introduction

Medulloblastoma (MB), a high-grade malignancy, is the most frequent brain tumor in children, accounting for 30% of all pediatric intracranial tumors and 7% to 8% of all brain tumors and has a 5-year survival rate of approximately 70–75% [1,2]. MB originates in the cerebellum or posterior fossa, arising from the fourth ventricle or vermis, and can spread throughout the brain and spine via cerebrospinal fluid [2,3]. Surgery, specifically maximal safe resection, was the sole treatment for MB tumors until the 1950s, when craniospinal irradiation became a widely accepted adjuvant treatment, followed by the acceptance of chemotherapy as another standard treatment modality in the 1990s [2]. Over the years, advanced research has led to improved diagnostic, surgical, and radiation technologies, resulting in improved patient outcomes. However, a successful treatment protocol ultimately depends on the patient’s risk stratification and tumor staging. 

Until recently, Chang’s staging has been a primary tool for MB risk stratification [2]. Initially incorporating tumor size and the presence of metastasis at the time of initial diagnosis to classify the tumor into stages, this stratification system has since incorporated other parameters such as age at diagnosis and amount of residual disease after surgery [2,4,5]. Based on these variables, MB patients are stratified into (1) high-risk: patients aged 3 or younger with a residual tumor greater than or equal to 1.5 cm^2^ following maximal safe resection and (2) average-risk: patients with none of the features associated with high-risk status. 

More recently, MB has been sub-categorized, based on cytogenetic profiles, to have molecular subgroups with distinct survival outcomes, which are named based on the cellular pathway activation they exhibit [6,7,8,9,10,11]. The 4 MB subgroups have been identified as Wingless (WNT), with a generally good prognosis; Sonic Hedgehog (SHH), which exhibits varied patient outcomes according to the age group; Group 3, with a poor prognosis; and Group 4, with an intermediate prognosis. Recent clinical trials are focusing on targeted therapies for each molecular subgroup to produce better personalized treatment plans [12]. Additionally, as long-term MB survivors have substantial neurologic and cognitive complications [4,13,14,15,16], some of these clinical trials have been primarily directed toward assessing the efficacy of therapy de-escalation in hopes of reducing long-term radiation-induced neurotoxicity in patients with average risk [17]. However, despite the advances in molecular sub-characterization of MB tumors, which have dramatically improved targeted therapies and patient outcomes, the prognosis for MB tumors remains inadequate. The significant tumor microenvironment heterogeneity reported across MB patients [10,18] results in distinct biologic behaviors that impact the disease classification as well as the treatment protocols [18]. 

Magnetic Resonance Imaging (MRI) (e.g., a gadolinium-based contrast agent (Gd-T_1w_), T2-w, T2-FLAIR sequences) remains the modality of choice for diagnosis, surgical guidance, and post-treatment follow-up response assessment in MB tumors. For instance, treatment response assessment in MB tumors requires tumor delineation to reliably compute measurements in two perpendicular planes (bidirectional or 2D) on clinical MRI scans, based on consensus recommendations by the response assessment in pediatric neuro-oncology (RAPNO) working group [19]. Recently, however, the field of *radiomics* (high-throughput extraction of large amounts of imaging features) [20] has provided a mechanism to exploit large qualifiable tumor-specific features on routine MRI scans that go beyond the sub-optimal bidirectional measurements. These radiomic features have been evaluated to capture tumor characteristics, including (a) semantic or qualitative features (i.e., radiologist-derived assessments of the tumor including speculations, size of the tumor along several axes) [20], (b) voxel-level texture features including gray-level features, which capture intra-tumoral heterogeneity measures [21], (c) morphometric (i.e., shape-based) features that quantitatively measure tumor boundary changes based on their 3D topology [22], as well as (d) tissue deformation features which capture the impact of tumor-related mass effect in the tumor microenvironment [21], on clinical MRI scans. These radiomic approaches have shown great promise in diagnostic, predictive modeling, and clinical decision-making [21,22,23,24,25,26] in the context of adult tumors (including brain tumors); however, they have only recently been explored for pediatric brain tumors. 

Similarly, radiogenomics, or imaging genomics, has served as a conduit to create “virtual biopsy” maps by establishing statistical associations of radiomic features with genomic and molecular information, including point mutations and signaling pathways of biological significance. With the increasing relevance of MB molecular subgroups, including SHH, WNT, Group 3, and Group 4 tumors, in modifying treatment decisions, there have been recent attempts to identify imaging/radiomic correlates of these molecular sub-types and their associations with survival in MB tumors [6,9,10]. These radiogenomic approaches may provide additional insights into a better understanding of tumor etiology and ultimately develop personalized, biologically relevant, and non-invasive treatment plans [27]. 

While there are excellent review papers focused on applications of radiomics and radiogenomics in adult tumors (including brain tumors) [21,22,23,24,25,26], these computational approaches have only recently been investigated for pediatric brain tumors. Thus, in this review article, we attempt to comprehensively and systematically compile a total of 15 existing articles that have investigated radiomic and radiogenomic approaches solely in the context of pediatric MB. Our objectives in this review article are three-fold. The first is to provide an overview of a typical radiomic and radiogenomic pipeline in the context of MB tumors. Secondly, we provide a detailed literature review of the radiomic and radiogenomic approaches that have thus far been explored for pediatric MB. We have grouped these studies by their endpoint, resulting in three main categories: (1) applications in survival prognostication [28,29,30,31,32], (2) applications in molecular subgroup classification [32,33,34,35,36,37,38,39,40] (i.e., radiogenomics), and (3) MB tumor segmentation [41,42]. Finally, we address the current challenges and future directions pertaining to applications of radiomic and radiogenomic approaches in pediatric MB tumors. Table 1 lists the acronyms used in this review paper.

## 2. Overview of Radiomic and Radiogenomic Pipelines

Figure 1 shows a typical radiomic/radiogenomics workflow. Typically, the workflow starts with data acquisition from the imaging scanners, followed by pre-processing operations that attempt to overcome scanner artifacts and data shifts across multi-institutional scans. In this context, many approaches have been developed for intensity standardization [32] or artifact removal (including motion artifacts) occurring during image acquisition, a common problem when acquiring scans from pediatric patients. Pre-processing is then followed by segmenting the region of interest (i.e., the tumor and its compartments of interest) from the imaging modalities [41,42,43,44,45]. Following tumor segmentation, radiomic analysis is performed, which involves extracting the different feature classes (e.g., texture, shape, size, structural deformations) from the tumor compartments [28,29,30,31,32,33,34,35,36,37,38,39,40]. Typically, the feature extraction stage is followed by some operations for pruning and reduction of the feature sets to remove redundant, highly correlated features. This can be conducted using different statistical approaches, such as logistic regression [29,31,32], minimum redundancy, maximum relevance [33], Pearson’s correlation coefficient [30,39], and principal component analysis [28]. The set of selected features is then fed into different machine learning and statistical models that pertain to a specific application. For instance, logistic regression models are employed for survival prognostication, such as Cox proportional hazard models and Least Absolute Shrinkage and Selection Operator (LASSO) regression [35,36,38,39]. Similarly, different machine learning classifiers may be employed for identifying and classifying the different molecular subgroups, such as Support Vector Machine (SVM)-based classifiers [33,34,35], Random Forest (RF)-based classifiers [37], etc. Performance metrics are finally employed to assess the model’s ability to perform the designated task, as well as its generalizability, to be applied in different clinical applications. There is a wide variety of performance metrics that could be included, from Area Under the Curve (AUC) [33,34,40], to Kaplan–Meier curves [28,29,30,31,32] for survival analysis, hazard ratios, etc. 

The radiogenomics pipeline builds on the radiomics pipeline by integrating the extracted radiomic features with the corresponding molecular information of the tumors, seeking associations across omics and imaging. This paired (imaging-omics) information is exploited to build comprehensive models that aim at guiding treatment strategies as well as evaluating patient outcomes. These models are based on understanding disease etiology and the biological underpinning of the disease.

## 3. Literature Review

In this section, we provide an overview of the existing literature for the developed radiomic and radiogenomic approaches in the context of pediatric MB tumors. Our search keywords for this review included a combination of keywords: “MB”, “radiomics”, “radiogenomics”, “molecular subgroup”, “survival prediction”, and “automated tumor segmentation”. This resulted in a total of 15 articles on PubMed and Google Scholar, which we have grouped by their endpoint, resulting in three main categories: (1) MB tumor segmentation [41,42] (2) applications in survival prognostication [28,29,30,31,32], (3) applications in molecular subgroup classification [32,33,34,35,36,37,38,39,40] (i.e., radiogenomics). Interestingly, our search yielded works in the literature that were largely focused on radiogenomics, including predicting molecular subgroups [32,33,34,35,36,37,38,39,40] rather than survival and predicting patient outcomes [28,29,30,31,32]. Below we present these radiomic and radiogenomic approaches as well as the major findings from these studies. 

### 3.1. Segmentation of Pediatric MB Tumors

Radiation treatment planning in MB tumors requires careful tumor delineation. Similarly, treatment response assessment in MB requires tumor delineation to reliably compute measurements in two perpendicular planes (bidirectional or 2D). Automated segmentation tools could substantially augment treatment planning in pediatric MB. Recently, deep-learning architectures, including Fully Convolutional Networks and U-Net [41,43,46,47], have allowed for the development of reliable and fully automated segmentation approaches for various types of solid tumors, including adult brain tumors [46]. These approaches focus on building fully convolutional encoder-decoder networks without fully connected layers to achieve end-to-end tumor segmentation [47]. However, deep learning has only recently been employed for the automated segmentation of pediatric brain tumors in a handful of studies [41,42,43,44,45]. The reported dice scores from the non-enhancing tumor, necrosis, and edema sub-compartments in these studies have been sub-optimal, which underlines the challenges with segmenting pediatric brain tumors. For instance, Peng et al. [42] developed a deep-learning network to automatically segment the tumors of high-grade gliomas, MB, and other leptomeningeal diseases in pediatric patients, on T1w contrast-enhanced and T2/FLAIR images. Similarly, the work in [41] employed a convolutional neural network (CNN)-based model to segment the sub-compartments of multiple pediatric brain tumors, primarily gliomas and included a limited cohort of MB cases (*n* = 24). The model processed images at multiple scales simultaneously using a dual pathway. The first pathway kept the images at their normal resolution, while the second pathway down-sampled them. While the model was able to differentiate between the enhancing and non-enhancing tumor compartments of MB tumors, the reported dice scores were relatively low (0.62 for enhancing tumor, 0.18 for edema, and 0.26 for non-enhancing tumor). It is generally noted that the area of pediatric MB tumor segmentation is understudied, and more automated, reliable approaches are needed on this front toward more effective radiation therapy planning in pediatric MB.

### 3.2. Survival Prognostication in Pediatric MB Using Radiomic Approaches

Table 2 summarizes the works conducted in the literature in the context of pediatric MB risk stratification. We provide a detailed summary of each of those approaches below. 

#### 3.2.1. Feature Extraction and Selection

The different approaches found in the literature in the context of MB risk stratification utilized a wide range of radiomic features on different MRI protocols, including T1w, CE-T1w, Gadolinium-enhanced T1w (Gd-T1w), T2w, FLAIR images, and perfusion imaging. For instance, Grist et al. [28] and Yan et al. [29] utilized the Apparent Diffusion Coefficient (ADC) maps from Dynamic Susceptibility Contrast (DSC) MRI perfusion images to predict survival in MB. Specifically, Grist et al. utilized the ADC maps along with T2w images and diffusion-weighted images (DWI) to extract imaging features from 17 MB cases. The feature set included statistics from the ADC maps (e.g., kurtosis, mean, etc.), mean of corrected Cerebral Blood Volume (CBV), and mean of uncorrected CBV (uCBV), in addition to the postoperative tumor volume to stratify patients into low- and high-risk groups. Similarly, Yan et al. [29] utilized the ADC maps, yet along with multiple MRI protocols, including T1w, CE-T1w, T2w, and FLAIR, to extract 5929 radiomic features (shape features, first-order intensity features, and higher-order texture features). Next, Intraclass Correlation Coefficient (ICC) was used for feature reduction before feeding them into the risk stratification statistical model. 

Liu et al. [30] and Zheng et al. [31] focused on routine MR imaging in their analysis. For instance, Liu et al. [30] constructed a radiomic model on multi-institutional data that comprised 253 MB pediatric patients, with a training cohort and two hold-out test sets. Specifically, a total of 1294 radiomic features were extracted from T1w images as well as contrast-enhanced T1w (CE-T1w) images (647 features from each modality) that include size, shape, and textural features. Feature selection was then conducted using Pearson’s correlation coefficient. Zheng et al. [31] constructed a radiomic model for risk stratification on a total of 111 children with pathologically confirmed MB. One thousand one hundred thirty-two radiomic features were extracted from CE-T1w images that include first-order statistics, volume, shape, gray-level co-occurrence matrix (GLCM), gray-level run-length matrix, gray-level size zone matrix, and gray-level dependence matrix. Feature reduction was then conducted using ICC.

Interestingly, Iyer et al. [32] explored radiomic features outside of the tumor that can help quantify the mass effect occurring in the healthy “brain around tumor” regions due to the tumor pushing and displacing the neighboring structures. Specifically, Gadolinium-enhanced T1-weighted (Gd-T1w) images of 88 MB patients were analyzed, where local tissue deformation heterogeneity features captured from the “brain around tumor” regions were extracted. These features were analyzed to identify differences between high-risk patients with highly heterogenous tumors and low-risk patients that have less heterogeneous tumors for the purpose of survival prediction.

#### 3.2.2. Statistical Models for Survival Prognostication

Most works for MB survival prediction have utilized logistic regression models and Cox proportional hazards models to risk-stratify MB patients. For instance, Grist et al. [28] utilized Cox regression analysis along with iterative Bayesian survival analysis to select the top features extracted from the ADC maps and the other modalities for survival prognostication. Additionally, both unsupervised machine learning (using K-means clustering) and supervised machine learning (using the Bayesian features with an RF classifier, a single-layer neural network, and an SVM classifier were employed for risk stratification with 10-fold cross-validation. The unsupervised clustering technique yielded an elevated Hazard Ratio (HR) of 5.6, confidence intervals of 1.6–20.1, and *p* < 0.001 for the high-risk patients. Applying supervised machine learning techniques that employed the Bayesian features combined with a single-layer neural network with 10-fold cross-validation provided an accuracy of 98% in risk stratification. 

Iyer et al. [32] constructed their survival model from the deformation heterogeneity deformation features, along with Chang’s stratification components for the MB subjects as well as their molecular subgroup information using multivariate logistic regression models. The radiomic deformation features yielded significant differences between low- and high-risk patients (*p* = $2.9\times 10^{-4}$, Concordance Index (CI) = 0.7). Interestingly, the deformation features combined with Chang’s classification and molecular stratification yielded the best results in risk-stratifying patients into low- and high-risk (*p* = 0.005, CI = 0.75). 

Liu et al. [30] employed Cox regression analysis and LASSO regression on their set of selected radiomic features to identify the features with the most prognostic value. A radiomic signature was constructed based on this set of features to predict progression-free survival (PFS) and overall survival (OS). Kaplan–Meier analysis and the log-rank test revealed that the prognostic model yielded C-indices of 0.71, 0.7, and 0.72 on the training and the hold-out test sets 1 and 2, respectively. Further, a radiomics nomogram that integrates the radiomic features, age, and metastasis was constructed and performed better than the nomogram incorporating only clinicopathological factors (C-index = 0.723 vs. 0.665 and 0.722 vs. 0.677 on the held-out test sets 1 and 2, respectively). Similarly, Yan et al. [29] employed LASSO regression and Cox proportional hazards regression for the identification of the top features for survival prediction. Clinicomolecular factors, comprising age, sex (female or male), Karnofsky Performance Status (KPS), molecular subgroups (WNT, SHH, Group 3 or Group 4), the extent of resection (complete or incomplete), radiation therapy (yes or no), and chemotherapy (yes or no) were also incorporated into the survival prediction models. The Wilcoxon test and chi-square test were used to assess differences in survival between the risk groups. Kaplan–Meier analysis, along with the log-rank test, revealed that the radiomics-clinicomolecular signature predicted OS (C-index = 0.762) and PFS (C-index = 0.697) better than either the radiomics signature (C-index: OS: 0.649; PFS: 0.593) or the clinicomolecular signature (C-index: for OS = 0.725; for PFS = 0.691) alone. 

Zheng et al. [31] also employed multivariate Cox regression and LASSO models to create a radiomic signature for risk stratification and obtain a radiomic score for each subject by using a linear combination of selected radiomics features and their weighted coefficients. Additionally, an integrative model combining radiomic features, clinical features, and conventional MRI features was constructed. The models were then evaluated using Kaplan–Meier analysis and C-indices. The radiomic features combined with clinical and conventional MRI features yielded the best results for predicting OS (C-index = 0.82) compared to using the radiomic signature alone (C-index = 0.7) in the training set. On the test set, C-indices were 0.78 and 0.75 using the integrative model and the radiomic model, respectively. This was observed in other works as well [29,30,32], where integrating the radiomic features with clinical and molecular parameters improves the performance of the risk stratification models rather than using any of these parameters alone. 

### 3.3. Molecular Subgroup Identification in Pediatric MB Using Radiomic Approaches

Table 3 summarizes the works conducted in the literature in the context of pediatric MB molecular subgroup classification. It was generally observed that combining radiomic features with clinical and demographic information generated the best results in terms of both survival prognostication and molecular subgroup identification. 

#### 3.3.1. Feature Extraction and Selection

In the context of identifying the 4 MB molecular subgroups, several models have utilized textural analysis on the tumor regions to identify differences between the subgroups. [33,34,35,36,37,38,39,40] For instance, Chang et al. [33] attempted to find the imaging surrogates of the 4 MB molecular subgroups using radiomic analysis in a study of 38 MB patients. Specifically, a total of 253 radiomic features that include tumor intensity, shape and size, and texture features were extracted from five different imaging sequences (T1w, T2w, FLAIR, ADC, and CE-T1w). This was followed by applying different feature selection algorithms, including minimum redundancy, maximum relevance, sequential backward elimination, and sequential forward selection to obtain the best future combination. Similarly, Iv et al. [34] developed a computational framework to predict the molecular subgroups of 109 MB patients collected from three different sites, where 590 radiomic features were extracted from T1w and T2w MR images. Namely, the features included intensity-based histograms, tumor edge-sharpness, Gabor features, and local area integral invariant features. A non-parametric Wilcoxon rank sum test was used for feature selection. Additionally, Saju et al. [35] employed texture analysis on the CE-T1w and T2w MR images of 38 MB patients, where features that included first- and second-order GLCM and shape features were extracted. Feature reduction was then conducted using LASSO regression.

In a similar fashion, Wang et al. [36] attempted to predict SHH and Group 4 subgroups on 95 MB patients (divided in the ratio of 7:3 for training: test sets) using their T1w, T2w, CE-T1w, and FLAIR sequences in addition to their ADC maps. Specifically, 7045 radiomic features that include intensity statistics, texture features that quantify the tumor heterogeneity (e.g., gray-level run-length and gray-level co-occurrence), shape and size, and high-order statistical features (using various filters such as exponential, logarithmic, square, square root, and wavelet) were extracted from the image sequences. This was followed by employing feature reduction algorithms to remove the redundant features, such as variance threshold, SelectKBest, and the LASSO regression model. Yan et al. [37] developed a radiomic model to predict the molecular subgroups of 122 MB subjects. Five thousand five hundred twenty-nine radiomic features were extracted from T1w, CE-T1w, T2w, and FLAIR MR images, in addition to the ADC maps of these patients. Namely, tumor location, shape features, intensity-based features, and texture features were extracted. The texture and intensity features were extracted from both the MR images and the transform-domain images. Feature pruning was then employed using ICC to remove the redundant features. Finally, Zhang et al. [38] constructed a model to identify the 4 molecular subgroups of 263 MB patients using their CE-T1w and T2w MR images. Specifically, 1800 radiomic textural features were extracted and then reduced using LASSO regression.

Aside from textural analysis, Iyer et al. [32] conducted a statistical approach to identify differences between the 4 MB subgroups on the Gd-T1w images of 71 patients. After extracting radiomic features that quantify the structural deformations occurring in the “brain around tumor” regions due to mass effect, this was followed by statistical analysis to classify the four subgroups. Also, Dasgupta et al. [39] conducted a study on 111 MB patients to predict their molecular subgroups from T1w, T2w, and diffusion imaging. Specifically, imaging features such as tumor location and size, diffusion characteristics, tumor margin, and T2w characteristics were extracted. A correlation between those individual features and the molecular subgroup was then established using statistical methods. 

Interestingly, a deep learning approach was previously adopted for the task of MB molecular stratification. Specifically, Chen et al. [40] developed a multi-tasked CNN-based approach that utilizes different information, including genotyping and prognosis, to predict the molecular subgroup of 113 MB patients. Using the tumor mask, this multi-staged model employed feature extraction from CE-T1w and T2w MR images using a ResNet model, region proposal, and subgroup prediction. The ResNet model used pyramid representations to construct feature pyramids, which were then used in the second stage to obtain region proposals that contain tumor lesions. Finally, each feature map of a region proposal was transformed into fixed spatial dimensions for the tasks of molecular subgroup prediction, prognosis, and tumor segmentation in a multi-task learning technique. In a 3-fold cross-validation scheme, the molecular subgroup prediction task, with the assistance of tumor segmentation and prognosis tasks, achieved AUCs of 0.96, 0.96, 0.99, and 0.96 for WNT, SHH, Group 3, and Group 4 subgroups, respectively. 

#### 3.3.2. Statistical Models for Molecular Subgroup Identification

Most of the approaches utilized in the context of MB molecular subgroup identification have employed logistic regression along with machine learning classifiers [33,34,35,36,37,38]. For instance, Chang et al. [33] implemented an SVM classifier with nested leave-one-out cross-validation (LOOCV) to find the best model from their extracted set of texture features. Based on the selected set of features (8 GLCM features), a prediction model was constructed, which generated (AUC) values of 0.82, 0.72, and 0.78 for WNT, Group 3, and Group 4, respectively. Similarly, IV et al. [34] employed an SVM classifier for the molecular subgroup prediction task using a cross-validation strategy. From their set of 590 radiomic features, the tumor edge-sharpness feature was found to be the most discriminative feature between SHH and Group 4 molecular subgroups. In this study, it is noted that the scans were acquired from different scanner vendors with different imaging parameters, which may affect the model’s performance metrics. In order to account for these variations, the authors have conducted extensive validation on their cohorts. Specifically, two predictive models were developed; one was based on a double 10-fold cross-validation scheme, where the subjects from the three datasets were combined, whereas the other model employed a three-dataset cross-validation strategy, where the model was trained using two datasets and tested on the third independent cohort. The 10-fold cross-validation model applied on the MRI modalities combined (T1w, T2w) yielded AUCs of 0.79, 0.7, and 0.83 for predicting SHH, Group 3, and Group 4 subgroups, respectively. Similarly, the 3-dataset cross-validation strategy resulted in predicting the SHH group with an AUC of 0.7–0.73 as well as Group 4 with an AUC of 0.76–0.8. In the work by Saju et al. [35], an SVM classifier was also employed in a 10-fold cross-validation strategy for model development. The authors used both One-vs-One and One-vs-All multiclass classification approaches for evaluation. Multiple models were sequentially evaluated by the system using a combination of the selected features to find the best predictive model. The best model was obtained by using a combination of 30 GLCM and six shape features on CE-T1w MR images. A 10-fold cross-validation demonstrated AUCs of 0.93, 0.9, 0.93, and 0.93 in predicting WNT, SHH, Group 3, and Group 4 MB subgroups, respectively. 

In the work by Wang et al. [36], based on the feature reduction step employed on the 7045 extracted features, a total of 17 optimal features were used to develop the classification model, which yielded classification accuracies with AUCs of 0.96 and 0.75 in the training and the test cohorts, respectively. Interestingly, when combining the radiomic features with the location of the tumor, the pathological type, and the hydrocephalus status of the two molecular subgroups, the model performance was improved, achieving AUCs of 0.965 and 0.849 in the training and the test cohorts, respectively. Yan et al. [37] also constructed a classification model which was RF-based and yielded 11 optimal features out of the 5529 extracted to predict the molecular subgroups. This model yielded AUCs of 0.83, 0.67, 0.6, and 0.7 for WNT, SHH, Group 3, and Group 4, respectively, for the test cohort of 30 patients. Further, incorporating tumor location and hydrocephalus status into the radiomic model improved the AUCs for WNT and SHH subgroups to 0.84 and 0.83, respectively. Finally, adding age and gender information to the model further improved the AUCs to 0.91 and 0.87 for WNT and SHH subgroups, respectively, and the classification accuracies for Group 3 and Group 4 were 70% and 86.67%, respectively. Uniquely, Zhang et al. [38] developed a two-stage model that comprises a binary classifier in each step for WNT, SHH, and non-WNT and non-SHH classes. The first stage was used to distinguish WNT and SHH from Group 3/Group 4 subgroups, whereas the second stage was used to distinguish WNT from SHH. Six different classifiers, namely, SVM, logistic regression, k-nearest neighbor, RF, extreme gradient boosting, and neural network, were employed in each stage, and the overall performance was assessed for the combined stages. The final multiclass classifier was guided by maximizing the Dice Coefficient (DC), calculated as the weighted average between precision and recall. The combined, sequential classifier achieved a DC score of 88% and a binary score of 95%, specifically for the WNT subgroup. Additionally, a Group 3 versus Group 4 classifier achieved an AUC of 98%. One of the notable limitations in this work is the heterogeneity across MRI scans that were collected from 12 different sites with various protocols and scanners. The authors attempted to mitigate this issue by performing z-score normalization to the images prior to feature extraction in order to improve the robustness of the radiomic model and the classification task.

Other statistical tests were employed in the context of MB molecular subgroup identification. For instance, Iyer et al. [32] utilized a multiclass ANOVA test, followed by multiple comparison of means, to identify significant differences between the four subgroups based on the deformation heterogeneity features extracted from the neighboring structures to the tumor. Significant differences were observed between deformation magnitudes obtained for Group 3, Group 4, and SHH subgroups that occurred up to 60 mm outside the tumor edge. The skewness of deformation yielded a *p*-value of 0.028 for Group 3 vs. SHH and Group 4, and the median of deformation yielded a *p*-value of 0.05 for Group 3 vs. Group 4. Similarly, Dasgupta et al. [39] applied some statistical tests such as the Pearson chi-square test, Fisher’s exact test, and Cohen’s Kappa statistics to establish a correlation between the imaging features and molecular subgroups. Additionally, on the training cohort (N = 76), binary logistic regression was performed using different combinations of the significant MRI features to distinguish a certain molecular subgroup from the other three, and nomograms were constructed for the individual subgroups. The predictive accuracies for the subgroups were excellent for SHH (95%), acceptable for Group 4 (78%), but sub-optimal for Group 3 (56%) and WNT (41%) subgroups. 

## 4. Challenges and Future Directions

From the compiled literature on the radiomic and radiogenomic approaches for MB pediatric brain tumors, we identified some common challenges and limitations, which we elaborate upon below.

### 4.1. Limited Sample Size and Class Imbalances

One reported limitation in most of the papers that we included in this review was the limited sample size, which can drastically affect the performance of computational models, including risks of overfitting and lack of independent validation. For instance, the literature focused on survival prognostication and employed sample sizes ranging between 17–253 subjects, with a median of 112 subjects. Similarly, for the papers focused on molecular subgroup stratification, the sample sizes ranged between 38–263, with a median of 109 subjects. One possible explanation for this limitation is the lack of availability of large pediatric brain tumor cohorts, as there are far more adults diagnosed with brain tumors than children (around 350 pediatric MB cases are diagnosed annually in the United States [48], compared to around 25,000 adults estimated to be diagnosed with brain tumors in 2023 [49]). One observation from the literature review we conducted is that the cohorts larger than 100 subjects are curated through multiple institutions, which can be a long-term direction toward solving the problem of limited data in radiomic analysis for MB studies. In addition, multiple clinical trials that include >200 MB pediatric subjects [17] may be made available in the coming years for testing and validating computational approaches. Further, utilizing the ongoing efforts with current consortiums that facilitate pediatric data curation, such as the Children’s Brain Tumor Network [50], is one of the solutions to the problem of limited cohorts. 

Another observation in the conducted literature review was the class imbalance with regard to the molecular subgroup categories. For instance, the curated cohorts in some studies [29,38] had a very limited sample size of the WNT subgroup compared to the other subgroups, which may have biased the performance of the radiomic models and hence decreased their sensitivity. Some of the recommended approaches to overcome this problem, among several others, are bootstrapping [38], sampling [51], and one-class learning [51]. 

### 4.2. Data-Shift and Model Generalizability across Multi-Institutional Studies

Curating studies retrospectively from multiple institutions is beneficial for evaluating the robustness and generalizability of the developed computational approaches. From a clinical standpoint, different scanners with different field strengths and different imaging acquisition parameters (e.g., manufacturer, field strength, spatial resolution, pulse acquisition parameters, scanner image filters) are acquired on a regular basis for pediatric tumor diagnosis. Hence, constructing radiomic models that account for and are robust to the image variations across sites and scanners will be paramount for demonstrating clinical utility [34]. Unfortunately, the variations introduced when employing multi-institutional data affect the reproducibility of radiomic features, in particular, the textural features which rely on per-voxel intensity measurements. Several works have come up with different pre-processing strategies to standardize MRI scans [32,35,37,38,52]. Some works also tend to assess the resilience of the radiomic features extracted from data acquired across different institutions by training their approaches on subjects curated from one institution and then performing external validation on an independent test cohort [34,40]. Others tend to train and test their models on data combined from different institutions [32,36] by performing random grouping. This strategy may allow for the incorporation of a range of differently acquired scans in the training model and hence increase the model’s generalizability. Several studies also tend to apply k-fold cross-validation schemes to evaluate the efficacy of their models on multi-institutional cohorts [34].

### 4.3. Lack of Uniformity in the Treatment Strategies across the Different MB Risk Groups

Non-uniform treatment protocols for MB patients have been reported in the literature [32], which may confound risk stratification analysis. Currently, the ongoing clinical trials for pediatric MB target specific risk groups, thus enrolling patients with uniform treatment regimens. With public access to clinical trial data, utilizing these cohorts of uniformly treated patients in radiomic and radiogenomic analysis in the future can afford opportunities to build more robust models that are not affected by modifications to initial treatment protocols.

### 4.4. Unavailability of Molecular Subgroup Information

Some studies have reported the unavailability of molecular subgroup information in their patients’ cohorts [30,36], which posed some restrictions on their analysis, such as lacking the ability to extend their algorithms to the identification of all molecular subgroups [36] or needing to perform an analysis on a subset of the available datasets for molecular subgroup identification [32]. 

### 4.5. Linking the Extracted Radiomic Features to the Underlying Disease Pathobiology

A future direction that still needs some effort while utilizing radiomics and radiogenomics in pediatric MB is to link the findings of these works with the biological underpinning of the disease and help understand the disease etiology. Some of the existing works have attempted to build nomograms and signatures to predict patient outcomes and risk-stratify patients [30,32,39]. However, there is still a need for additional studies in this area to obtain biological validation of the findings and thus have better clinical decision-making and improved outcomes for the pediatric cancer community. 

## 5. Conclusions

The works reviewed in this paper demonstrate the promise of applying radiomics and radiogenomics for advancements in diagnosis, prognostication, and improving patient outcomes in pediatric medulloblastoma. However, there are still challenges in this area, prominently a lack of publicly available studies for computational analysis. However, with the recently launched clinical trials and with data sharing efforts across different institutions as well as data consortiums, this challenge might be mitigated in the coming years, ultimately allowing for building reliable and reproducible machine learning models trained on large multi-institutional cohorts.

## Figures and Tables

**Figure 1 diagnostics-13-02727-f001:**
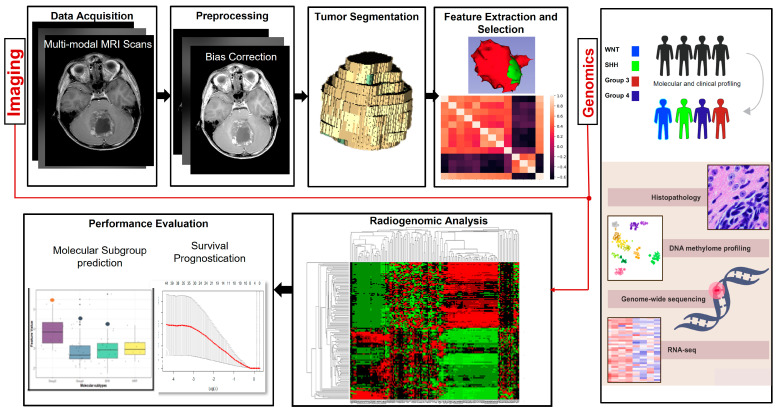
Typical radiomic workflow for radiomic and radiogenomic approaches employed for pediatric medulloblastoma research.

**Table 1 diagnostics-13-02727-t001:** List of acronyms used in this paper.

Acronym	Term	Acronym	Term
ADC	Apparent Diffusion Coefficient	ICC	Intraclass Correlation Coefficient
AUC	Area Under the Curve	KPS	Karnofsky Performance Status
CBV	Cerebral Blood Volume	LASSO	Least Absolute Shrinkage and Selection Operator
CE	Contrast-Enhanced	LOOCV	Leave-One-Out Cross-Validation
CI	Concordance Index	MB	Medulloblastoma
CNN	Convolutional Neural Network	MRI	Magnetic Resonance Imaging
DC	Dice Coefficient	OS	Overall Survival
DSC	Dynamic Susceptibility Contrast	PFS	Progression-Free survival
DWI	Diffusion-Weighted Images	RF	Random Forest
Gd-T1w	Gadolinium-enhanced T1-weighted	SHH	Sonic Hedgehog
GLCM	Gray-level co-occurrence matrix	SVM	Support Vector Machine
HR	Hazard Ratio	uCBV	Uncorrected Cerebral Blood Volume
		WNT	Wingless

**Table 2 diagnostics-13-02727-t002:** Summary of previous works utilizing radiomic approaches for MB survival prognostication.

Group	Radiomic Endpoint	Sample Size	Single or Multi-Institution	Mean Age (Years) Unless Otherwise Denoted	Features	Modality	Models & Feature Selection Methods	Performance Metrics/Statistical Analysis	Model Performance	Limitations
Grist et al., 2021 [28]	Survival prediction	17	Multi	8.85	ADC maps (kurtosis, mean, etc.), mean of corrected CBV, mean of uncorrected CBV, tumor volume	T2w, DWI, DSC	Cox regression; Iterative Bayesian analysis; KNN; SVM; RF	Kaplan–Meier Analysis, HR	Unsupervised clustering: HR = 5.6, confidence intervals = 1.6–20.1, *p* < 0.001 for high-risk patients Supervised machine learning: Bayesian features with a single-layer neural network & 10-fold cross-validation provided 98% accuracy	Small cohort size
Iyer et al., 2022 [32]	Survival prediction	88 (*n* = 60 for training, *n* = 28 for testing)	Multi	5.4	Deformation heterogeneity features	Gd-enhanced T1w	Logistic regression, Cox models, LASSO	Kaplan–Meier Analysis, HR, CI	$\mathrm{Deformation}\;\mathrm{features}\;\mathrm{yielded}\;p=2.9\times 10^{-4}$, CI = 0.7 between low- and high-risk patients Deformation features with Chang’s and molecular stratification yielded best results in risk-stratifying patients (p=0.005, CI = 0.75)	Small cohort size Lack of uniformity in the treatment strategies for the risk groups
Liu et al., 2021 [30]	Survival prediction	253 (113: training; 113:hold-out test set 1; 27:hold-out test set 2)	Multi	7.4 for training set; 8.1 for hold-out test set 1; 6.8 for hold-out test set 2	647 features per modality (8 size and shape, 639 texture)	T1w, CE-T1w	Pearson’s correlation, Cox Regression with LASSO	Kaplan–Meier Analysis, Kruskal–Wallis test	Predictive model of PFS yielded C-indices of 0.71, 0.7, and 0.72 on training and hold-out test sets 1 and 2. The radiomics nomogram integrating radiomic features, age, metastasis performed better than the nomogram incorporating clinicopathological factors (CI = 0.723 vs. 0.665 and 0.722 vs. 0.677 on the held-out test sets 1 and 2)	Molecular information was not involved. Limited size for hold-out test set 2
Yan et al., 2020 [29]	Survival prediction	166 (83: training,83: testing)	Single	Median: 8	5929 features (shape, first-order intensity, higher-order texture).	T1w, CE-T1w, T2w, FLAIR, ADC maps	ICC, LASSO, Cox regression	Kaplan–Meier Analysis; Wilcoxon test/chi-square test	Radiomics-clinicomolecular signature predicted OS (CI = 0.762), PFS (CI = 0.697) better than radiomics signature (CIs = 0.649,0.593 for OS, PFS) or the clinicomolecular signature (CIs = 0.725, 0.691 for OS, PFS)	Limited sample size Lack of volumetric MRI data
Zheng et al., 2022 [31]	Survival prediction	111 (77: training, 34: testing)	Single	5.82	1132 features (first-order statistics, volume, shape, GLCM, gray-level run-length matrix, gray-level size zone matrix)	CE-T1w	Cox regression model, LASSO	T-test, Mann–Whitney U test, Fisher’s exact/chi-square test	Radiomic features + clinical + conventional MRI features yielded best results for predicting OS (CI = 0.82), vs. using the radiomic signature alone (CI = 0.7) on training set CIs were 0.78 and 0.75 using the integrative model and the radiomic model, on the test set	Limited sample size Data was from a single institution. Molecular information was not available

**Table 3 diagnostics-13-02727-t003:** Summary of previous works utilizing radiomic approaches for identification of MB molecular subgroups.

Group	Radiomic Endpoint	Sample Size	Single or Multi-Institution	Mean * Age (Years) * Unless Otherwise Denoted	Features	Modality	Models & Feature Selection Methods	Performance Metrics/Statistical Analysis	Model Performance	Limitations
Chang et al., 2021 [33]	Molecular classification	38 (WNT: 7, SHH: 12, Group 3: 8, Group 4: 11)	Multi	7.5	253 features (intensity, shape and size, texture)	T1w, T2w, FLAIR, CE- T1w, ADC	minimum redundancy maximum relevance; sequential backward elimination; SVM	Accuracy, Sensitivity, Specificity	The model based on 8 GLCM features has AUCs of 0.82, 0.72, and 0.78 for WNT, Group 3, and Group 4	Limited sample size
Iyer et al., 2022 [32]	Molecular classification	71 (*n* = 49 for training- WNT:4, SHH:15, Group 3:8, Group 4: 22; *n* = 22 for testing- WNT:3, SHH:6, Group 3:3, Group 4:10)	Multi	5.4	Deformation heterogeneity features	Gd-T1w	Multiclass ANOVA; multiple comparison of means	HR, CI	*p*-values = 0.028 for Group 3 vs. SHH and Group 4, 0.05 for Group 3 vs. Group 4	Small cohort size Lack of uniformity in the treatment strategies for the different subgroups Mutation information for the molecular subgroups was not available
Chen et al., 2020 [40]	Molecular classification	113 (*n* = 74 for validation- WNT:17, SHH:18, Group 3:20, Group 4:19; *n* = 39 for testing- WNT:7, SHH:9, Group 3:11, Group 4: 12)	Multi	4.4 for infants, 10.5 for children	Feature pyramid network & refined feature layers of Residual neural network (ResNet101)	CE-T1w, T2w	Mask-RCNN model: feature extraction, region proposal, prediction.	Kruskal–Wallis test, AUC, sensitivity, specificity	Accuracy of 0.93 in the cross-validation cohort and 0.85 in the testing cohort. AUCs of molecular subgrouping were 0.97 and 0.92 in cross-validation and independent test cohorts	Limited sample size No information about evidence of spinal metastasis to predict dissemination
Dasgupta et al., 2019 [39]	Molecular classification	111 (WNT: 17, SHH: 44, Group 3: 27, Group 4: 23)	Multi	Median = 9	Tumor size, MR Imaging characteristics	T1w, T2w, DWI	Multimodal logistic regression, nomogram construction	Pearson chi-square test, Fisher’s exact test, Cohen’s Kappa statistics	Overall molecular subgroup accuracy = 74%; 95% SHH, 78% Group 4, 56% Group 3, 41% WNT	No reliable prediction of WNT and Group 3 A uniform MRI protocol was not used No correlation between magnetic resonance spectroscopy findings & molecular subgrouping
IV et al., 2019 [34]	Molecular classification	109 (WNT: 19, SHH: 30, Group 3: 24, Group 4: 36)	Multi	8.7 (across three sites)	590 features (intensity-based histograms, tumor edge-sharpness, Gabor, local area integral invariant features)	T1w, T2w	Wilcoxon rank sum test, SVM classifier	AUC, ROC curves	Double 10-fold cross-validation for predicting SHH, Group 3, Group 4 using combined T1w and T2w images yielded AUCs = 0.79, 0.70, and 0.83, respectively	Heterogeneity in image data (different scanners, etc.) Limited imaging sequences
Saju et al., 2022 [35]	Molecular classification	38 (WNT:7, SHH:7, Group 3:12, Group 4:12)	Single	Median = 9	82 features from each modality; first and second-order GLCM and shape features	CE-T1w, T2w	LASSO, SVM	AUC, ROC curves	10-fold cross-validation yielded AUCs of 0.93, 0.9, 0.93, and 0.93 in predicting WNT, SHH, Group 3, and Group 4	Very limited sample size
Wang et al., 2023 [36]	SHH and Group 4 prediction	95 (SHH:47, Group 4:48; ratio 7:3 training: test)	Multi	6.75 for SHH, 7.5 for Group 4	7045 features (intensity statistics, texture, shape and size, high-order statistics)	T1-, CE- T1-, T2-weighted, FLAIR, ADC	LASSO	T-test, Fisher’s exact test, Delong test, AUC, ROC curves	Classification model with 17 optimal features yielded AUCs of 0.96 and 0,75 in training and test cohorts	Limited sample size No external validation No inclusion of WNT, Group 3
Yan et al., 2020 [37]	Molecular classification	122 (92 for training- WNT:15, SHH:16, Group 3:40, Group 4:21; 30 for testing- WNT:6, SHH:4, Group 3:14, Group 4:6)	Single	11.57	5929 features (location, shape, intensity, texture)	T1w, CE-T1w, T2w, FLAIR, ADC	ICC, RF-based wrapper algorithm, logistic regression	Kruskal–Wallis test, Wilcoxon test, ROC, AUC	Incorporating tumor location, gender, age, and hydrocephalus with radiomics generated AUCs of 0.91 and 0.86 for WNT and SHH	Advanced MR sequences not included Limited sample size Nanostring assay was utilized for molecular subgrouping, which is not a calibrated assay
Zhang et al., 2022 [38]	Molecular classification	263 (WNT: 26, SHH: 83, Group 3/4: 154; 75:25 for training: test set)	Multi	10.1 for WNT, 6.9 for SHH, 12.8 for Group 3/4	1800 texture features	CE-T1w, T2w	Binary classifier along with SVM, logistic regression, KNN, RF, extreme gradient boosting, neural network	Wald test, Dice Similarity Score	Combined, the sequential classifier achieved a DC score of 88% and a binary score of 95% for WNT. Group 3 vs. Group 4 classifier achieved an AUC of 98%	Limited sample size Heterogeneity of MR scans (12 sites) Features extracted from isolated tumor volumes No incorporation of tumor-brain spatial relationships

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
