# Peer review of "Opportunities and Advances in Radiomics and Radiogenomics for Pediatric Medulloblastoma Tumors"

_diagnostics, 2023, doi:10.3390/diagnostics13172727_

Round 1

Reviewer 1 Report

This is a well-structured manuscript that comprehensively summarizes the current data on radiomics/radiogenomics in pediatric medulloblastoma.

Minor comments: 

Table 2 and 3:  although quite informative, they could be graphically optimized

Line 196:  “in addition  to the tumor volume to stratify patients into low- and high-risk groups” should it rather read “in addition  to the postoperative tumor volume to stratify patients into low- and high-risk groups” ?

Author Response

We thank the reviewers for their feedback. We appreciate the enthusiastic comments of the reviewers on the merits of our work such as “well-structured”, “much-needed contribution”, “well-written and insightful”. In this rebuttal, we substantially addressed all the concerns of the reviewers and the editor and have included the details of this rebuttal in the revised manuscript.

Reviewer: 1

  1. Table 2 and 3:  although quite informative, they could be graphically optimized.

We have now worked on optimizing the tables (2, 3) by cutting down the text to make it easier to read. Fonts in the tables have also been made smaller to make it less crowded and look neat. The tables now include more concise information that highlights the takeaways from each paper as well as breaks down the cohorts analyzed.

  1. Line 196: “In addition to the tumor volume to stratify patients into low- and high-risk groups” should it rather read “in addition to the postoperative tumor volume to stratify patients into low- and high-risk groups”?

We thank the reviewer for the excellent point. We have now rewritten the sentence (page 7 in the revised manuscript) to reflect that the measures of the postoperative tumor volume are the ones employed in survival analysis.

Reviewer 2 Report

This review article is a much-needed contribution, focusing on the radiomics and radiogenomics of paediatric medulloblastoma tumours. The article is well-written and insightful.

However, there are several criticisms that I would like to address:

1)      The issue of generalizability is a critical aspect in any biomarker study. While the review mentions this concern, I recommend providing a more comprehensive discussion and emphasizing its significance throughout the text. Ideally, an entire subsection should be dedicated to the generalizability  issue. It would also be beneficial to include specific details about the size of the training, holdout, and external groups in each of the 15 studies covered. Additionally, if available, insights on whether there is an observable trend indicating that a higher number of cross-validation folds during model selection enhances generalizability would be of high interest to readers.

2)      The tables included in the article seem overly crowded with extensive text, making them challenging to follow. This congestion might be due to issues during the Word-to-PDF conversion process. I advise reducing the font size within the tables and condensing the descriptions to improve readability.

Author Response

We thank the reviewers for their feedback. We appreciate the enthusiastic comments of the reviewers on the merits of our work such as “well-structured”, “much-needed contribution”, “well-written and insightful”. In this rebuttal, we substantially addressed all the concerns of the reviewers and the editor and have included the details of this rebuttal in the revised manuscript.

  1. The issue of generalizability is a critical aspect in any biomarker study. While the review mentions this concern, I recommend providing a more comprehensive discussion and emphasizing its significance throughout the text. Ideally, an entire subsection should be dedicated to the generalizability issue. It would also be beneficial to include specific details about the size of the training, holdout, and external groups in each of the 15 studies covered. Additionally, if available, insights on whether there is an observable trend indicating that a higher number of cross-validation folds during model selection enhances generalizability would be of high interest to readers.

We thank the reviewer for raising this excellent point. We have now revisited the works that were exposed to the issue of curating data from multiple cohorts and how they tackled the issue of data heterogeneity while building their radiomic models. Any validation schemes they adopted in their work to generalize the models and increase their robustness are now addressed in detail (Section 3.3.2). Additionally, all the sizes of the training/validation/test cohorts are listed in Tables 2, 3 for all the works explored in this review.  In Section 4.2, we further elaborate on the issue of data-shift and model generalizability; we introduce the issue and its challenges as well as possible ways to optimize the radiomic models’ performance and robustness while analyzing datasets coming from different institutions.

b) The tables included in the article seem overly crowded with extensive text, making them challenging to follow. This congestion might be due to issues during the Word-to-PDF conversion process. I advise reducing the font size within the tables and condensing the descriptions to improve readability.

We have now worked on optimizing the tables (2, 3) by cutting down the text to make it easier to read. Fonts in the tables have also been made smaller to make it less crowded and look neat. The tables now include more concise information that highlights the takeaways from each paper as well as breaks down the cohorts analyzed.